# Revisiting Discriminator in GAN Compression: A Generator-discriminator Cooperative Compression Scheme

**Shaojie Li**[1],[*] **Jie Wu**[2], **Xuefeng Xiao**[2], **Fei Chao**[1], **Xudong Mao**[1], **Rongrong Ji**[1,3],[†]

[1]Media Analysis and Computing Lab, Department of Artificial Intelligence,
School of Informatics, Xiamen University
[2] ByteDance Inc [3]Institute of Artificial Intelligence, Xiamen University

## Abstract

Recently, a series of algorithms have been explored for GAN compression, which aims to reduce tremendous computational overhead and memory usages when deploying GANs on resource-constrained edge devices. However, most of the existing GAN compression work only focuses on how to compress the generator, while fails to take the discriminator into account. In this work, we revisit the role of discriminator in GAN compression and design a novel generator-discriminator cooperative compression scheme for GAN compression, termed GCC. Within GCC, a selective activation discriminator automatically selects and activates convolutional channels according to a local capacity constraint and a global coordination constraint, which help maintain the Nash equilibrium with the lightweight generator during the adversarial training and avoid mode collapse. The original generator and discriminator are also optimized from scratch, to play as a teacher model to progressively refine the pruned generator and the selective activation discriminator. A novel online collaborative distillation scheme is designed to take full advantage of the intermediate feature of the teacher generator and discriminator to further boost the performance of the lightweight generator. Extensive experiments on various GAN-based generation tasks demonstrate the effectiveness and generalization of GCC. Among them, GCC contributes to reducing 80% computational costs while maintains comparable performance in image translation tasks. Our code and models are available at: `https://github.com/SJLeo/GCC`

## 1 Introduction

Generative Adversarial Networks (GANs) have been widely popularized in diverse image synthesis tasks, such as image generation [16, 50], image translation [64, 30] and super resolution [35, 59]. However, the ultra-high computational and memory overhead of GANs hinder its deployment on resource-constrained edge devices. To alleviate this issue, a series of traditional model compression algorithms, i.e., network pruning [18, 36], weight quantization [39, 26], low-rank decomposition [10, 19], knowledge distillation [22, 52], and efficient architecture design [29, 55] have been applied to reduce calculational consumptions of GANs.

Previous work [56, 12, 61] attempted to directly employ network pruning methods to compress the generator but obtain unimpressive results, as shown in Figure 1(a). The similar phenomenon also occurred in SAGAN shown in Appendix A. A potential reason is that these methods failed to take into account the generator and the discriminator must follow the Nash equilibrium state to avoid the mode

---

[*]This work was done when Shaojie Li was a intern at Bytedance Inc.
[†]Corresponding Author: rrji@xmu.edu.cn

35th Conference on Neural Information Processing Systems (NeurIPS 2021).

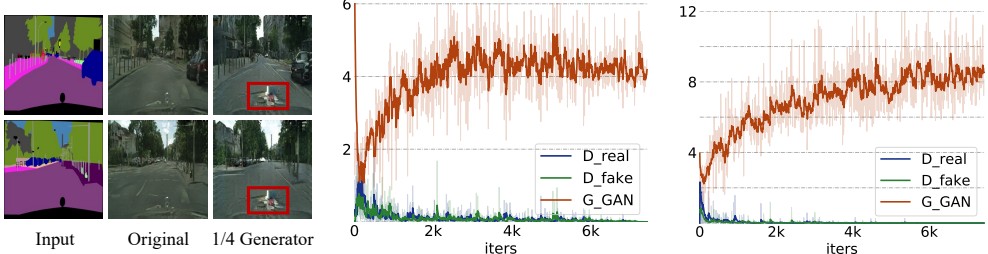

(a) Visualization of mode collapse    (b) Original generator + original    (c) 1/4 channels width generator +
discriminator             original discriminator

Figure 1: Illustration of model collapse phenomenon. The experiment is conducted on Pix2Pix [30] based on the Cityscapes [8] dataset. (a) shows the influence of mode collapse in generating images. Each image generated by the 1/4 channels width generator appears the forgery trace, which is marked by the red box. (b) and (c) show loss curves of the original generator and the 1/4 channels width generator with the original discriminator, respectively.

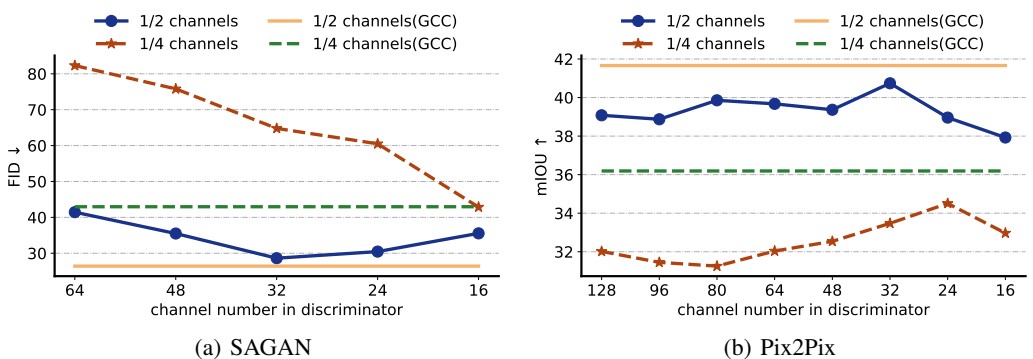

(a) SAGAN                                   (b) Pix2Pix

Figure 2: The performance of generators with different compression ratios on various channel number discriminators. (a) is the result of SAGAN [63] on CelebA [44] dataset. (b) is the result of Pix2Pix on the Cityscapes dataset. "1/2 channels" means that we reduce the network width of the generator network to 1/2 of the original size. GCC uses selective activation discriminator without distillation.

collapse in the adversarial learning. In other words, these methods simply compress the generator while maintaining the original capability of the discriminator, which resulted in breaking the balance in adversarial learning. As shown in Figure 1(b) and 1(c), when compressing the generator, the loss of the discriminator gradually tends to zero, such a situation indicates that the capacity of the discriminator significantly surpasses that of the lightweight generator. Furthermore, the capacity imbalance between the generator and discriminator leads to the mode collapse problem [47].

In order to retain the Nash equilibrium between the lightweight generator and discriminator, we must revisit the role of the discriminator in the procedure of GAN compression. A straightforward method is to reduce the capacity of the discriminator to re-maintain the Nash equilibrium between it and the lightweight generator. Therefore, we train a lightweight generator against the discriminators with different channel numbers. The results are shown in Figure 2, we can obtain the following observations: i) The channel numbers in the discriminator significantly influence the performance of the lightweight generator; ii) The discriminator with the same channel compression ratio as the generator may not get the best result. iii) The optimal value of channel numbers for discriminators is task-specific. In short, it is arduous to choose an appropriate channel number for the discriminator in different tasks.

To solve the above issues, we design a generator-discriminator cooperative compression scheme, term GCC in this paper. GCC consists of four sub-networks, i.e., the original uncompressed generator and discriminator, a pruned generator, and a selective activation discriminator. GCC selectively activates

partial convolutional channels of the selective activation discriminator, which guarantees the whole optimization process to maintains in the Nash equilibrium stage. In addition, the compression of the generator will damage its generating ability, and even affect the diversity of generated images, resulting in the occurrence of mode collapse. GCC employs an online collaborative distillation that combines the intermediate features of both the original uncompressed generator and discriminator and then distills them to the pruned generator. The introduction of online collaborative distillation boosts the lightweight generator performance for effective compression. The contributions of GCC can be summarized as follows:

- To the best of our knowledge, we offer the first attempt to revisit the role of discriminator in GAN compression. We propose a novel selective activation discriminator to automatically select and activate the convolutional channels according to a local capacity constraint and a global coordination constraint. The dynamic equilibrium relationship between the original models provides a guide to choose the activated channels in the discriminator.

- A novel online collaborative distillation is designed to simultaneously employ intermediate features of teacher generator and discriminator to guide the optimization process of the lightweight generator step by step. These auxiliary intermediate features provide more complementary information into the lightweight generator for generating high-fidelity images.

- Extensive experiments on various GAN-based generation tasks (i.e., image generation, image-to-image translation, and super-resolution) demonstrate the effectiveness and generalization of GCC. GCC contributes to reducing 80% computational costs while maintains comparable performance in image translation.

## 2   Related Work

**Generative Adversarial Networks.**   Generative Adversarial Network, i.e., GANs [16] has attracted intense attention and has been extensively studied for a long time. For example, DCGAN [50] greatly enhanced the capability of GANs by introducing the convolutional layer. The recent works [17, 33, 63, 4] employed a series of advanced network architecture to further improve the fidelity of the generated images. In addition, several novel loss functions [54, 17, 2, 45] were designed to stabilize the adversarial training stage of GANs. A popular research direction of GANs is image-to-image translation. The image-to-image translation is designed to transfer images from the source domain to the target domain while retaining the content representation of the image. Image-to-image translation is widely used in computer vision and image processing, such as semantic image synthesis [30, 48], style transfer [64, 1], super resolution [35, 59], image painting [49, 65] and so on.

**Mode Collapse.**   Mode collapse is the main catastrophic problem during the adversarial training stage of GANs, which leads to the diversity of generated images are monotonous, only fitting part of the real data distribution. To address this issue, recent efforts were focus on new objective functions [47, 17, 34], architecture modifications [5, 15] and mini-batch discrimination [54, 33]. Actually, the capacity gap between the generator and the discriminator is a more essential reason for mode collapse. In this paper, we provide a different perspective to eliminate the capacity gap, we select the convolutional channel to be activated in the discriminator according to a local capacity constraint and a global coordination constraint.

**GAN Compression.**   To reduce huge resource consumption during the inference process of GANs, great efforts have emerged in the field of GAN compression in the recent two years. The search-based methods [56, 37, 12, 23, 40, 58] exploited neural architecture search and genetic algorithm to obtain a lightweight generator. However, they fell into the trap of manual pre-defined search space and huge search costs. The prune-based methods [38, 31, 57, 61] directly pruned a lightweight generator architecture from the original generator architecture. However, these works failed to take discriminator pruning into account, which would seriously destroy the Nash equilibrium between the generator and the discriminator. The discriminator-free methods [51, 12] directly use the large GAN model as a teacher to distill the lightweight generator without discriminator, which also achieved good performance. Slimmable GAN [23] correspondingly shrank the network width of the discriminator with the generator. However, as shown in Fig. 2, there is no linear between the channel numbers of the optimal discriminator and generator. Anycost GAN [40] proposed a generator-conditioned discriminator, which generates the discriminator architecture via passing the generator architecture

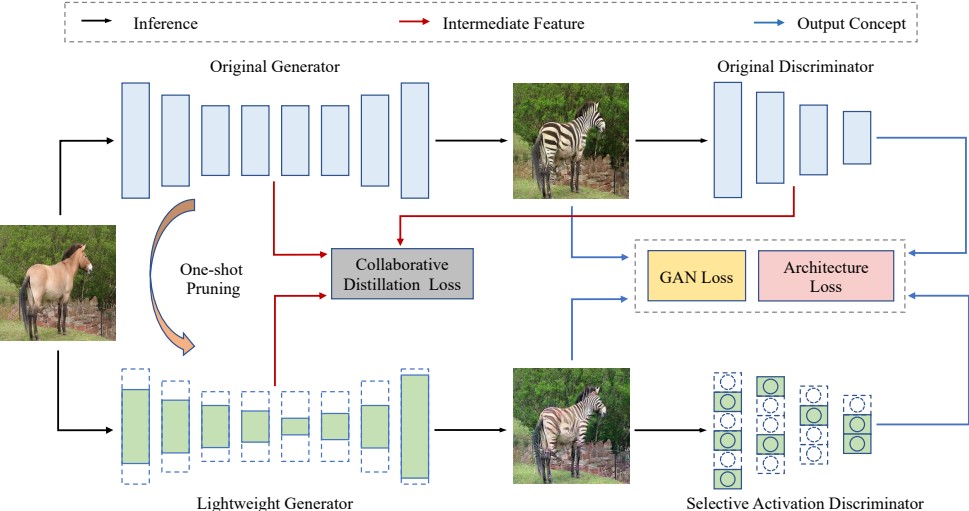

Figure 3: The proposed generator-discriminator cooperative compression scheme (GCC).

into the fully connected layer. However, the fully connected layer is only trained through the original GAN loss, so it is difficult to accurately predict the optimal discriminator architecture for the lightweight generator. In this work, we design a generator-discriminator cooperative compression (GCC) scheme, which employ a selective activation discriminator and novel online collaborative distillation to boost the performance of the lightweight generator.

## 3 Methodology

This section, we illustrate the proposed GCC framework in Figure 3. We first briefly introduce the composition and loss function of GAN in Section 3.1. Then, we present how to obtain the lightweight architecture from the original generator in Section 3.2. The selective activation discriminator that automatically chooses activated convolutional channels is illustrated in Section 3.3. In Section 3.4, we propose a novel online collaborative distillation scheme to further improve the performance of the lightweight generator.

### 3.1 Preliminaries

Generative Adversarial Networks (GANs) contain two fundamental components, generator $G$ and discriminator $D$. Among them, $G$ maps the input $\boldsymbol{z}$ into a fake image for cheating the discriminator, while the discriminator distinguishes the generator outputs from the real images $\boldsymbol{x}$. The generator and discriminator are alternately optimized via the adversarial term to achieve their respective optimization objectives. The optimization objectives are defined as follows:

$$\mathcal{L}_G = \mathbb{E}_{\boldsymbol{z} \sim p(\boldsymbol{z})} \left[ f_G(-D(G(\boldsymbol{z}))) \right] \tag{1}$$

$$\mathcal{L}_D = \underbrace{\mathbb{E}_{\boldsymbol{x} \sim p_{\text{data}}} \left[ f_D(-D(\boldsymbol{x})) \right]}_{\mathcal{L}_{D_{real}}} + \underbrace{\mathbb{E}_{\boldsymbol{z} \sim p(\boldsymbol{z})} \left[ f_D(D(G(\boldsymbol{z}))) \right]}_{\mathcal{L}_{D_{fake}}} \tag{2}$$

where $f_G$ and $f_D$ are the loss functions of the generator and discriminator, respectively. Theoretically, the generator and discriminator maintain the Nash equilibrium when the generator and discriminator have a balanced capacity, and the discriminator is cheated. In this paper, we observe from experiments that the Nash equilibrium maintained in the GAN compression stage to avoid model collapse and obtain an impressive performance.

### 3.2 Generator Compression

In order to avoid additional computational overhead brought by the neural architecture search, we directly employ the conventional network pruning methods to compress the generator. Previous

work [43] revealed that the core of network pruning searches for a suitable sub-architecture from the original network architecture instead of parameter inheritance. Therefore, we leverage the one-shot pruning method [36, 42] to obtain the appropriate lightweight generator architecture from the original generator architecture. In our experiments, we adopt the slimming [42] to prune the generator when the network contains the Batch Normalization (BN) layers. We use L1-norm pruning [36] on the contrary (without BN layers). The detailed pruning process is described in Appendix B. Finally, we only keep the architecture of the pruned generator and train it from scratch with a selective activation discriminator proposed in Section 3.3.

### 3.3 Selective Activation Discriminator

As shown in Figure 2, the convolutional channel number in the discriminator have an essential impact on the performance of the lightweight generator. Therefore, we design a selective activation discriminator to automatically choose and activate convolutional channels via learnable retention factors. We assign a learnable retention factor to each convolution kernel of the original discriminator, each retention factor represents the probability that the corresponding convolution kernel participates in the inference process. When the retention probability is larger than the given threshold, the corresponding convolution kernel is retained, otherwise it is suppressed. We denote retention factors set as $\mathbf{A} = \{\alpha_i\}_{i=1}^{L_D}$ where $\alpha_i = \{\alpha_{i1}, \alpha_{i2}, ..., \alpha_{in_i}\} \in \mathbb{R}^{n_i}$ is the retention factors of the $i$-th convolution layer and $n_i$ is the number of convolution kernel in the $i$-th layer. Given the threshold $\tau$, the retention factor $\alpha_{ij}$ determines whether the $j$-th convolution kernel of the $i$-th layer activate or not:

$$I_{ij} = \begin{cases} 1 & \text{if } \alpha_{ij} \in [\tau, 1] \\ 0 & \text{if } \alpha_{ij} \in [0, \tau). \end{cases} \tag{3}$$

Then activated convolution kernel's output $O'_{ij}$ is calculated by:

$$O'_{ij} = I_{ij} * O_{ij}, \tag{4}$$

where $O_{ij}$ is the original output feature map. Since the binary function is not differentiable for $\alpha_{ij}$, we employ STE [27] to estimate the gradient of $\alpha_{ij}$ as:

$$\frac{\partial \mathcal{L}_{arch}}{\partial \alpha_{ij}} = \frac{\partial \mathcal{L}_{arch}}{\partial I_{ij}}, \tag{5}$$

where $\mathcal{L}_{arch}$ is a loss function, and the retention factor $\alpha_{ij}$ is manually clipped to the range of $[0, 1]$. The learning of retention factor is considered as the architecture search of the discriminator.

The selective activation discriminator is optimized according to a local capacity constraint and a global coordination constraint. On the one hand, the incremental loss gap between the generator and the discriminator will break the Nash equilibrium in the adversarial training scheme, which leads to the discriminator effortlessly distinguishing the authenticity of the input images. The pruned generator and the selective activation discriminator should pursue capacity balance to maintain the Nash equilibrium. Therefore, a local capacity constraint $\mathcal{L}_{local}$ is designed as follows:

$$\mathcal{L}_{local} = |\mathcal{L}_G^S - \mathcal{L}_{D_{fake}}^S|, \tag{6}$$

where the superscript $S$ denotes the student (compressed model). On the other hand, the loss gap between the original model's combination internally reflects the relative capacity relationship between them. Therefore, we design a global coordination constraint to adjust the relative ability of the student generator and discriminator to be consistent with that of the teacher combination. This constraint helps to pull the loss gap between teacher and student models into close proximity. In this way, the uncompressed model is regarded as teacher model to guide the learning of retention factors to automatically adjust the discriminator capacity to match the lightweight generator. The global coordination constraint $\mathcal{L}_{global}$ is calculated by:

$$\mathcal{L}_{global} = |\mathcal{L}_{local} - |\mathcal{L}_G^T - \mathcal{L}_{D_{fake}}^T|| \tag{7}$$

The superscript $T$ represents the teacher (uncompressed) model. Due to the instability of adversarial training, we use Exponential Moving Average (EMA) to stabilize the loss corresponds to the teacher models. In short, the optimization objective of the retention factor is defined as follows:

$$\mathcal{L}_{arch} = \mathcal{L}_D^S + \mathcal{L}_{global}. \tag{8}$$

where $\mathcal{L}_D^S$ guarantees the basic identification function of the discriminator. Furthermore, we use the following bilevel optimization [7, 11] to optimize $\alpha$: i) Update discriminator weights to minimize $\mathcal{L}_D$ for $t$ times. ii) Update retention factor $\alpha$ to minimize $\mathcal{L}_{arch}$. iii) Repeat step i) and step ii) until the end of training. In the optimization process of the discriminator weights, we freeze the retention factors and vice versa.

### 3.4 Online Collaborative Distillation

In Section 3.3, we use the original model as the teacher models to guide the learning of the student discriminator. Most of the existing work merely considered the distillation from the generator, in contrast, we also employ richer intermediate information from the discriminator to perform collaborative distillation, which provides complementary and auxiliary concepts to boost the generation performance. Therefore, we propose a novel online collaborative distillation method to take full advantage of the teacher generator and discriminator combinations to promote the lightweight generator. The online scheme optimizes the teacher model and the lightweight student model iteratively and progressively from scratch. Therefore, we do not need a pre-trained teacher model, and directly complete the distillation process in one stage. Experiments show that online knowledge distillation is more efficient than traditional two-stage offline knowledge distillation.

Perceptual loss [32] is widely used to calculate the similarity of semantic and perceptual quality between images. It measures the semantic differences via a pre-trained image classification network (such as VGG), where the extracted features are less relevant to our generation task. Meanwhile, the teacher discriminator has a task-oriented ability to distinguish the authenticity of images. Following the work [6], we employ the teacher discriminator to replace the pre-trained classification model to extract the features of the images for calculating the perceptual loss. Furthermore, the intermediate feature map of the generator has rich information for generating real-like fake images, and we also add them to the calculation of perceptual loss. In this way, the similarity metric is summarized as:

$$d(\cdot, \cdot) = \gamma_m MSE(\cdot, \cdot) + \gamma_t Texture(\cdot, \cdot), \tag{9}$$

where $\gamma_m$ and $\gamma_t$ are pre-defined hyperparameter used for balancing MSE and Texture [13, 14] loss functions, respectively. Detailed formulation of Texture loss function is described in Appendix C. Thus, the collaborative distillation loss is:

$$\mathcal{L}_{\text{distill}} = \sum_{i=1}^{L_G} \mathbb{E}_{\boldsymbol{z} \sim p(\boldsymbol{z})} \left[ d(f_i(G_i^S(\boldsymbol{z})), G_i^T(\boldsymbol{z})) \right] + \sum_{j=1}^{L_D} \mathbb{E}_{\boldsymbol{z} \sim p(\boldsymbol{z})} \left[ d(D_j^T(G^S(\boldsymbol{z})), D_j^T(G^T(\boldsymbol{z}))) \right],$$
$$\tag{10}$$

where $G_i^S$ and $G_i^T$ are the intermediate feature maps of the $i$-th chosen layer in the student and teacher generator, $D_j^T$ are the intermediate feature maps of the $j$-th chosen layer in the teacher discriminator. $L_G$ / $L_D$ denote the number of the generator / discriminator chosen layers. $f_i$ is a $1 \times 1$ learnable convolution transform layer to match the channel dimension of the teacher's feature maps.

## 4 Experiments

### 4.1 Setups

To demonstrate the effectiveness of the proposed GCC, we conduct extensive experiments on a series of GAN based tasks:

**Image Generation.** We employ SAGAN [63] to input random noise to generate face images on the CelebA [44] dataset. CelebA contains 202,599 celebrity images with a resolution of $178 \times 218$. We first center crop them to $160 \times 160$ and then resize them to $64 \times 64$ before feeding them into the model. Frechet Inception Distance (FID) [21] is adopted to evaluate the quality of the generated images. The lower score indicates the better the image generation results.

**Image-to-Image Translation.** Pix2Pix [30] is a conditional GAN that uses Unet-based [53] generator to translate paired images. We evaluate pix2pix on the Cityscapes [8] dataset, which contains 3475 German street scenes. The images are resized to $256 \times 256$ for training. We follow the work [37] run the DRN-D-105 [62] to calculate mean Intersection over Union (mIOU) for evaluating the quality of the generated images. The higher score indicates the better the image generation performance. In addition, we also conduct experiments on the unpaired dataset. CycleGAN [64] leverages a

Table 1: Quantitative comparison with the state-of-the-art GAN compression methods.

| Model | Dataset | Method | MACs | Compression Ratio | Metric | |
|-------|---------|--------|------|-------------------|--------|--------|
| | | | | | FID($\downarrow$) | mIoU($\uparrow$) |
| SAGAN | CelebA | Original | 23.45M | - | 24.87 | - |
| | | Prune | 15.45M | 34.12% | 36.60 | - |
| | | GCC (Ours) | 15.45M | 34.12% | **25.21** | - |
| CycleGAN | Horse2zebra | Original | 56.80G | - | 61.53 | - |
| | | Co-Evolution [56] | 13.40G | 76.41% | 96.15 | - |
| | | GAN-Slimning [57] | 11.25G | 80.19% | 86.09 | - |
| | | AutoGAN [12] | 6.39G | 88.75% | 83.60 | - |
| | | GAN Compression [37] | 2.67G | 95.30% | 64.95 | - |
| | | CF-GAN [58] | 2.65G | 95.33% | 62.31 | - |
| | | CAT [31] | 2.55G | 95.51% | 60.18 | - |
| | | DMAD [38] | 2.41G | 95.76% | 62.96 | - |
| | | Prune | 2.40G | 95.77% | 145.1 | - |
| | | GCC (Ours) | 2.40G | 95.77% | **59.31** | - |
| Pix2Pix | Cityscapes | Original | 18.6G | - | - | 42.71 |
| | | GAN Compression [37] | 5.66G | 69.57% | - | 40.77 |
| | | CF-GAN [58] | 5.62G | 69.78% | - | 42.24 |
| | | CAT [31] | 5.57G | 70.05% | - | 42.53 |
| | | DMAD [38] | 3.96G | 78.71% | - | 40.53 |
| | | Prune | 3.09G | 82.39% | - | 38.12 |
| | | GCC (Ours) | 3.09G | 82.39% | - | **42.88** |

Table 2: Quantitative results on the super resolution task.

| Model | MACs | Set5 | | Set14 | | BSD100 | | Urban100 | |
|-------|------|------|------|-------|------|--------|------|----------|------|
| | | PSNR | SSIM | PSNR | SSIM | PSNR | SSIM | PSNR | SSIM |
| SRGAN | 145.88G | 29.88 | 0.86 | 27.07 | 0.76 | 26.35 | 0.72 | 24.77 | 0.76 |
| GCC(Ours) | 22.79G | **30.35** | **0.87** | **27.46** | **0.77** | **26.58** | 0.72 | **24.88** | 0.76 |

ResNet [20] generator to translates horse image into zebra image on the horse2zebra dataset extracted from ImageNet [9]. This dataset consists of 1,187 horse images and 1,474 zebra images, and all images are resized $256\times256$ for training. We use FID to evaluate the quality of the generated images.

**Super Resolution.** We apply the proposed GCC to compress SRGAN [35], which uses a DenseNet-like [24] generator to upscale low-resolution images. The training set and validation set of COCO [41] dataset are combined for the training process. We randomly crop $96\times96$ high-resolution images from the original images and then downsample it $4\times$ into low-resolution images as the input of the generator. For testing, we employ Set5 [3], Set14 [35], BSD100 [46], and Urban100 [25] as the standard benchmark datasets. Peak Signal-to-Noise Ratio (PSNR) [28] and Structural Similarity (SSIM) [60] are used to evaluate the quality of the generated images. The higher score indicates the better the image generation performance. More implementation details are illustrated in Appendix D.

### 4.2 Comparison to the State of the Art

**Quantitative Result.** Tables 1 and 2 show the comparison results of different algorithms on image generation, image-to-image translation, and super-resolution. From these two tables, we can obtain the following observations: 1) In the task of image translation, our method can greatly reduce the computational costs in each task and achieve comparable performance as the original uncompressed model. For example, GCC obtains a lower FID than the original model under the extremely high compression ratio on CycleGAN. Specifically, GCC reduces the MACs of CycleGAN from 56.8G to 2.40G, with compression by $23.6\times$ while FID still reduces 2.22. 2) GCC reduces the computational costs by 82.39% and achieves a higher mIoU than the original model, which establishes a new state-of-the-art performance for the Pix2Pix model. 3) In image generation task, GCC also shows impressive results on low MACs demands of GAN, i.e., SAGAN. Although the original SAGAN only requires 23.45M MACs to generate a $64\times64$ pixel image, GCC successes to reduce the MACs by 34.12%. 4) For SRGAN in the super-resolution task, GCC helps to significantly reduce 123.09G

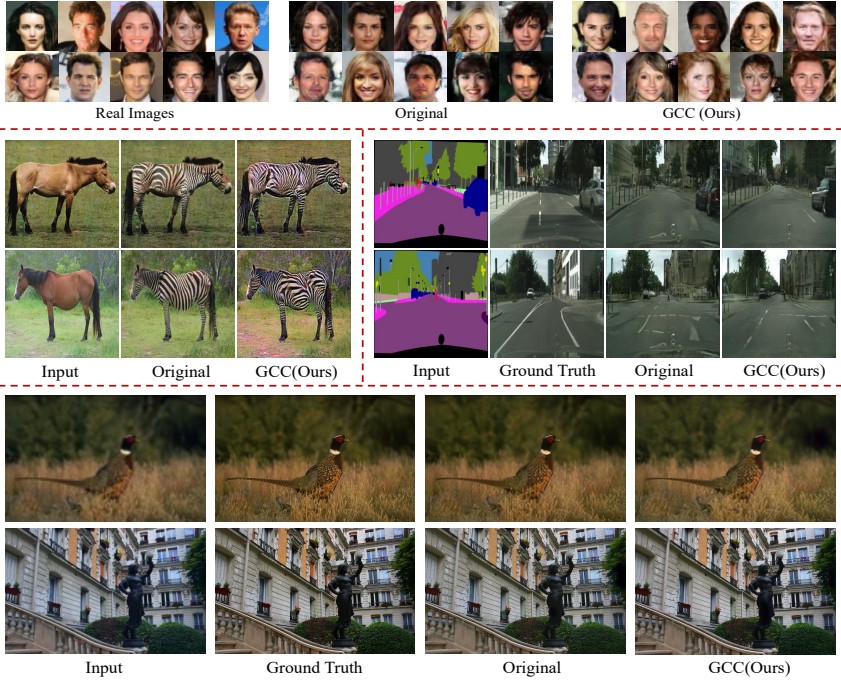

Figure 4: Qualitative results of SAGAN based on CelebA dataset (top), CycleGAN based on Horse2zebra dataset (middle left), Pix2Pix based on Cityscapes dataset (middle right), and SRGAN based on BSD100 / Urban100 dataset (bottom). Original represents the images generated by the original uncompressed generator. (Best viewed with zooming in)

MACs, with a compression ratio of $6.4\times$. It is worth noting that although GCC reduces such a high amount of computational costs, the lightweight model can achieve higher PSNR and SSIM than the original model on each test dataset. In addition, we also show the results of employing network pruning methods [36, 42] to compress the generator and directly perform adversarial training with the original discriminator, abbreviated as "Prune", the experimental results demonstrate the importance and effectiveness of our proposed selective activation discriminator. In summary, Our proposed GCC contributes to removes a large number of redundant calculations while maintains the comparable performances of the original model. The results provide a feasible solution for demands on a super-computing or mini-computing GAN model.

**Qualitative Results.** Figure 4 depicts the model generation performance on each task. Qualitative results reveal that GCC contributes to generating high-fidelity images with low computational demands. More qualitative results are provided in Appendix G.

**Acceleration.** We also compare acceleration performance on actual hardware between the original generator and the lightweight generator on Pix2Pix. The latency is measured on two types of computing processors (*i.e.* Intel Xeon Platinum 8260 CPUs and NVIDIA Tesla T4 GPUs). The original generator takes 75.96ms to generate one image on CPUs and 2.06ms on GPUs. However, the lightweight generator only uses 18.68ms and 1.74ms to generate one image on CPUs and GPUs, respectively. GCC helps to achieve $4.07\times$ and $1.18\times$ acceleration on CPU and GPU, respectively.

### 4.3 Ablation Study

**Online Collaborative Distillation.** Table 3 shows the effectiveness of each components in the online collaborative distillation. The variant "Ours w/o Online" represents the traditional two-stage distillation scheme. "Ours w/o D-distillation" and "Ours w/o G-distillation" denotes that only adopt the intermediate feature information of the teacher generator/discriminator in the distillation stage, respectively. We can intuitively observe that the distillation method can indeed further improve the performance of the model. Both MSE loss and Texture loss are helpful in the distillation process, and the combination of these two losses achieves better mIOU scores. GCC deploys the distillation process

Table 3: Ablation study results in online collaborative distillation.

| Name | Online | Collaborative | MSE loss | Texture loss | mIOU(↑) |
|------|--------|---------------|----------|--------------|---------|
| Ours w/o distillation | × | × | × | × | 39.17 |
| Ours w/o Texture loss | ✓ | ✓ | ✓ | × | 41.80 |
| Ours w/o MSE loss | ✓ | ✓ | × | ✓ | 40.07 |
| Ours w/o Online | × | ✓ | ✓ | ✓ | 41.16 |
| Ours w/o D-distillation | ✓ | × | ✓ | ✓ | 42.31 |
| Ours w/o G-distillation | ✓ | × | ✓ | ✓ | 41.43 |
| GCC(Ours) | ✓ | ✓ | ✓ | ✓ | **42.88** |

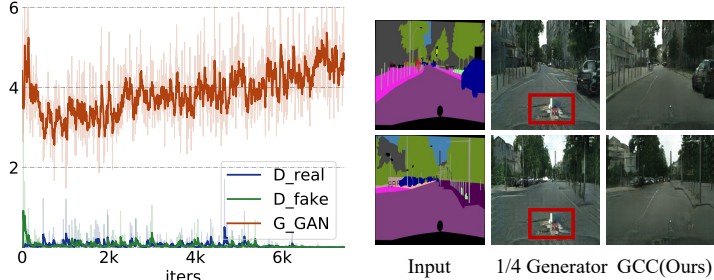

(a) 1/4 channels width generator with GCC  (b) Comparison of visualization results

Figure 5: Loss curve and visualization results via GCC improvement. The experiment was conducted on Pix2Pix based on the Cityscapes dataset. (Best viewed with zooming in)

into the online stage, where promotes an uncompressed teacher model progressively to gradually guide the learning of the lightweight generator and avoid the complicated two-stage distillation scheme. In Table 3, the online distillation scheme can achieve better performance than the offline distillation. The online collaborative distillation distills the intermediate feature information from both the generator and discriminator of the teacher model. G-only distillation / D-only distillation only distills one of them to the lightweight generator, which leads to declining results.

**Solution of Mode Collapse.** Figure 1 shows the phenomenon of mode collapse when compressing the generator without GCC. With the introduction of GCC, the 1/4 channels width generator can get rid of mode collapse. We show the improved loss curve and visualization results in Figure 5. In Figure 5(a), the selective activation discriminator effectively balances the adversarial training with the lightweight generator. The loss gap between the 1/4 channels width generator and selective activation discriminator is more evenly matched to that of the original model as shown in Figure 1(b). In Figure 5(b), the 1/4 channels width generator can generate higher quality images with GCC.

## 5   Conclusion

During GAN compression, we observe that merely compressing the generator while retaining the original discriminator destroys the Nash equilibrium between them, which further results in mode collapse. In this work, we revisit the role of discriminator in GAN compression and design a novel generator-discriminator cooperative compression (GCC) scheme. To ensure normal adversarial training between the lightweight generator and discriminator, we propose a selective activation discriminator to automatically select the convolutional channel to be activated. In addition, we propose a novel online collaborative distillation that simultaneously utilizes the intermediate feature of the teacher generator and discriminator to boost the performance of the lightweight generator. We have conducted experiments on various generation tasks, and the experimental results verify the effectiveness of our proposed GCC.

# 6 Acknowledge

This work is supported by the National Science Fund for Distinguished Young Scholars (No.62025603), the National Natural Science Foundation of China (No.U1705262, No.62072386, No.62072387, No.62072389, No.62002305, No.61772443, No.61802324 and No.61702136) and Guangdong Basic and Applied Basic Research Foundation (No.2019B1515120049).

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
