# Revisiting Discriminator in GAN Compression: A Generator-discriminator Cooperative Compression Scheme (Appendix)

## A  Mode Collapse in SAGAN

We also observe that the phenomenon of mode collapse on SAGAN. As shown in Figure 1(b) that merely compressing the generator and retraining the original discriminator will cause obvious loss oscillation. Similarly, as shown in the middle part of Figure II, the generated results are not impressive. However, the loss curve is much stable and the quality of the generated images are greatly improved with the introduction of our proposed GCC.

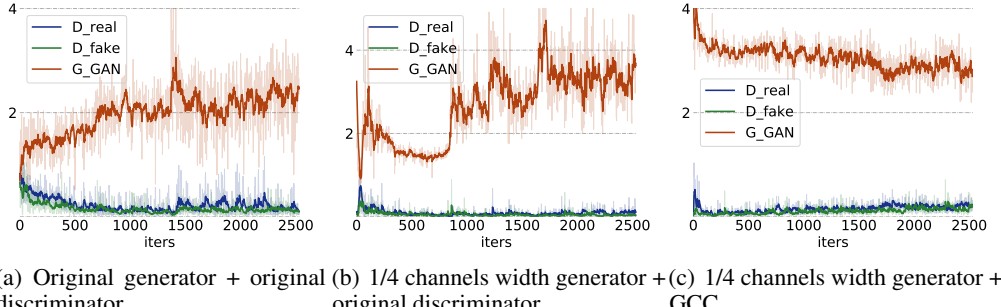

(a) Original generator + original discriminator

(b) 1/4 channels width generator + original discriminator

(c) 1/4 channels width generator + GCC

Figure I: Loss curves under different training settings. The experiment is conducted on SAGAN based on the CelebA dataset. (a), (b) and (c) show loss curves of the original generator, the 1/4 channels width generator with the original discriminator, and the 1/4 channels width generator with our proposed GCC, respectively.

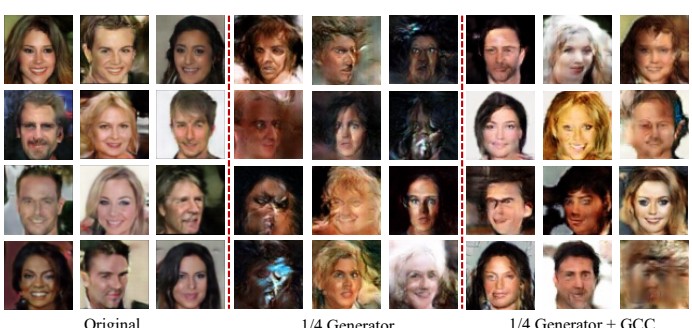

Original       1/4 Generator       1/4 Generator + GCC

Figure II: Illustration of model collapse phenomenon.

35th Conference on Neural Information Processing Systems (NeurIPS 2021).

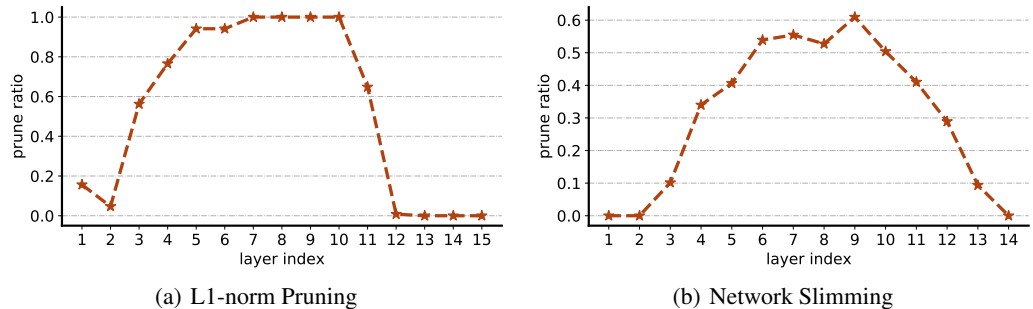

(a) L1-norm Pruning             (b) Network Slimming

Figure III: The pruning rate of Pix2Pix model based on Cityscapes dataset after L1-norm pruning or slimming pruning.

## B   Pruning Details

Network slimming [3] adds L1 regularization to the scale parameter in the BN layer, and finally employs the absolute value of the scale parameter as the important metric of the corresponding convolution kernel. L1-norm pruning [2] also adds L1 regularization to the weights of all convolution kernels and then uses the L1-norm of the convolution kernel as its importance metric. To avoid the impact of adding additional L1 regularization on the performance of the generator and additional fine-tuning after pruning. We only use 1/10 of the original training epoch to add the L1 regularization to the generator, and then sort all the convolution kernels in ascending order according to their importance metric. Given computational constrain, the pruning method removes the convolution kernel with less importance until the requirements are met.

To verify the adaptability of the pruning methods [2, 3] in the generator network, we visualize the pruning ratio of each layer in the Pix2Pix generator in Figure III. We find that the pruning ratio of each layer likes an inverted letter "U" as the number of network layers deepens. Pix2Pix generator takes U-Net as the backbone network, so the first/last few layers play a direct role in generating real-like fake images. The output feature map size is only 1×1 in the most intermediate network layer, which has little effect on the final 256×256 image. In conclusion, the pruning methods [2, 3] can better identify the important convolution kernel of the generator.

## C   Formulation of Texture loss function.

The purpose of Texture loss is to ensure that the two images have a similar style (*e.g.*, colors, textures, contrast). We directly use it to measure the similarity between two features. The feature similarity is regarded as the correlations between different feature channels and defined as the Gram matrix $G(O^l) \in \mathbb{R}^{c_l \times c_l}$, where $c_l$ represents the number of channels in the $l$-th layer output feature map $O^l$. We denote $G_{ij}(O)$ as the inner product between the $i$-th channel feature map of $O^l$ and $j$-th channel feature map of $O$. Then the texture loss function is calculated as follows:

$$\text{Texture}(\hat{O}, O) = \frac{1}{c_l^2} \sqrt{\sum_{i,j} \left( G_{ij}(\hat{O}) - G_{ij}(O) \right)^2} \tag{I}$$

where $\hat{O}$ and $O$ respectively represent different output feature maps.

# D   Implementation Details

Table I: Hyperparameter settings in the experiment.

| Model | Dataset | Training Epochs | | Batch Size | $\gamma_m$ | $\gamma_t$ | GAN Loss | ngf | | ndf |
| | | Const | Decay | | | | | Teacher | Student | |
|---|---|---|---|---|---|---|---|---|---|---|
| SAGAN | CelebA | 100 | 0 | 64 | 1 | 100 | Hinge | 64 | 48 | 64 |
| CycleGAN | Horse2zebra | 100 | 100 | 1 | 0.01 | 1e3 | LSGAN | 64 | 24 | 64 |
| Pix2Pix | Cityscapes | 100 | 150 | 1 | 50 | 1e4 | Hinge | 64 | 32 | 128 |
| SRGAN | COCO | 15 | 15 | 16 | 0.1 | 0.1 | Vanilla | 64 | 24 | 64 |

We use Pytorch to implement the proposed GCC on NVIDIA V100 GPU. We have conducted experiments on SAGAN[1], CycleGAN[2], Pix2Pix[3] and SRGAN[4] respectively, and hyperparameter settings are shown in Tab. I.

**Selective Activation Discriminator.** We optimize $\alpha$ via the ADAM optimizer with an initial learning rate of 0.0001, and decay by 0.1 every 100 epoch. The threshold $\tau$ of SAGAN, CycleGAN, Pix2Pix and SRGAN in Eq. 3 are set to 0.1, 0.1, 0.5, 0.1 respectively. In addition, we denote $|\mathcal{L}_G^T - \mathcal{L}_{D_{fake}}^T|$ of Eq.7 as $\mathcal{L}_{target}$. We use Exponential Moving Average (EMA) to stabilize $\mathcal{L}_{target}$ during the training process. The specific update strategy is as follows:

$$\mathcal{L}_{target} = \beta_t * \mathcal{L}_{target}^{t-1} + (1.0 - \beta_t) * \mathcal{L}_{target}^t \tag{II}$$

$$\beta_t = Epoch_{current} / Epoch_{total} \tag{III}$$

where t represents the current number of iterations. $Epoch_{current}$ / $Epoch_{total}$ represent the current / total epoch number of training.

Table II: Selection of distillation convolutional layer location. 'Total Number' represents the number of all convolution layers in the network, and 'Selected Number' represents the serial number of the selected convolutional layer.

| Model | Dataset | Generator Position | | Discriminator Position | |
| | | Total Number | Selected Number | Total Number | Selected Number |
|---|---|---|---|---|---|
| SAGAN | CelebA | 5 | 2, 4 | 5 | 2, 4 |
| CycleGAN | Horse2zebra | 24 | 3, 9, 15, 21 | 5 | 2, 4 |
| Pix2Pix | Cityscapes | 16 | 2, 4, 12, 14 | 5 | 2, 4 |
| SRGAN | COCO | 37 | 9, 17, 25, 33 | 4 | 2, 4 |

**Distillation Layers.** We show the position of the selected distillation layer in each network in Tab. II. We usually choose the nonlinear activation layer after the selected convolutional layer, otherwise we choose the normalization layer or the convolutional layer itself.

To sum up, we summarize the proposed GCC framework in Algorithm 1.

# E   Additional Ablation Study

**Selective Activation Discriminator.** We report the ablative studies of selective activation discriminator in Table III. Motivated by discriminator-free method [1], we discard the student discriminator and train the lightweight generator only with $L_{distill}$, which obtains 35.50 mIOU. It may be due to

---

[1]SAGAN repository: https://github.com/heykeetae/Self-Attention-GAN

[2]CycleGAN repository: https://github.com/junyanz/pytorch-CycleGAN-and-pix2pix

[3]Pix2Pix repository: https://github.com/junyanz/pytorch-CycleGAN-and-pix2pix

[4]SRGAN repository: https://github.com/sgrvinod/a-PyTorch-Tutorial-to-Super-Resolution

**Algorithm 1** The Proposed GCC Framework

**Input:** inputs $Z = \{z\}_i^N$, real images $X = \{x\}_i^N$, training epochs $E$, uncompressed generator $G^T$ and discriminator $D^T$, selective activation discriminator $D^S$, and retention factor $\alpha$.

**Output:** efficient lightweight generator.
1: *# First Step: Generator Compression*
2: **for** epoch = 1 : E / 10 **do**
3:     Update $G^T$ and $D^T$ using Eq. 1 and Eq. 2 respectively, and L1 regularization is added to the BN scale parameter or weight of $G^T$.
4: **end for**
5: Prune $G^T$ to obtain a lightweight generator $G^S$, and reinitialize $G^T$, $D^T$ and $G^S$.
6: *# Second Step: Lightweight Generator Training*
7: **for** epoch = 1 : E **do**
8:     Get a batch of $z_1$ from $Z$ and $x_1$ from $X$
9:     Update $G^T$ and $D^T$ using Eq. 1 and Eq. 2 respectively.
10:     Update $G^S$ using the combination of Eq. 1 and Eq. 10.
11:     Freeze $\alpha$ and train $D^S$ using Eq. 2.
12:     Get a batch of $z_2$ from $Z$ and $x_2$ from $X$
13:     Freeze $D^S$'s weight parameter and update $\alpha$ using Eq. 8.
14: **end for**

Table III: Ablation study results in selective activation discriminator.

| Name | Discriminator | Selective Activation | $L_{global}$ | mIOU |
|---|---|---|---|---|
| Ours w/o discriminator | × | × | × | 35.50 |
| Ours w/o selective activation | ✓ | × | ✓ | 37.24 |
| Ours w/o $L_{global}$ | ✓ | ✓ | × | 40.21 |
| GCC(Ours) | ✓ | ✓ | ✓ | **42.88** |

the fact that although the discriminator-free method avoids the model collapse issue, it fails to take good advantage of GAN loss to provide more supervision information for the lightweight generator. In order to investigate whether selective activation discriminator and global coordination constraint $L_{global}$ can work synergistically, we discard one of them and the experimental results indicate that the selective activation discriminator and $L_{global}$ work mutually to achieve impressive results.

## F  Border Impact

Our proposed GCC needs a teacher model whose generator can conduct normal adversarial training with a discriminator. The teacher model guides the learning of selective activation discriminator and the lightweight generator. Therefore, GCC relies on a good teacher model to ensure the effectiveness of compression. In addition, due to the instability of adversarial training, GAN may generate results that distort objective facts. This may be potential negative social impacts of our work.

## G  More Qualitative Results

We show more qualitative results in Figure IV, V, VI, and VII, respectively.

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

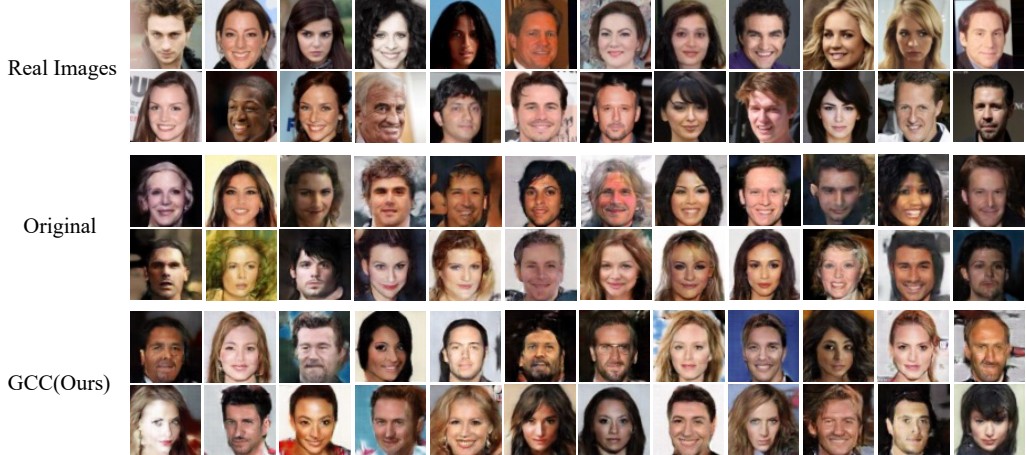

Figure IV: Qualitative results of SAGAN based on CelebA dataset.

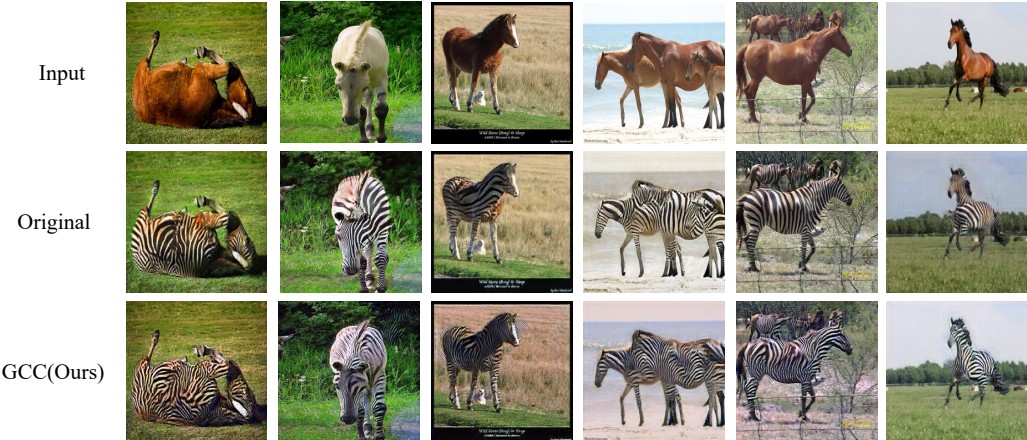

Figure V: Qualitative results of CycleGAN based on Horse2zebra dataset.