# OpenReview forum: "Revisiting Discriminator in GAN Compression: A Generator-discriminator Cooperative Compression Scheme"
_NeurIPS.cc/2021/Conference — NeurIPS 2021 Poster_

### Official Review · Reviewer_aeqa · 2021-07-14

**Rating:** 6
**Confidence:** 4

**Summary:**

In this paper, the authors revisit the role of discriminator in GAN compression and design a new generator-discriminator cooperative scheme GCC for GAN compression. Specifically, they use a selective activation discriminator to automatically select the convolutional channels, aiming to help maintain the Nash equilibrium and avoid mode collapse. Besides, they also propose an online collaborative distillation method to further boost the compression performance. They achieve impressive performance on three widely-used GAN tasks--image generation, image-to-image translation, and super-resolution.

**Limitations And Societal Impact:**

I think the authors have adequately addressed the limitations and potential negative societal impact of their work.


**Main Review:**

* Originality: The novelty is somewhat limited. As the authors mention in the related work, Slimmable GAN also tries to maintain the Nash equilibrium by making the discriminator width proportional to the generator width. I think the authors are just applying some NAS methods to automatically prune the discriminator, which is quite similar to Anycost GAN. Although the authors do a good job analyzing the correspondence of different generator and discriminator sizes, the basic idea behind it is not interesting.

* Quality:

  * I am curious about why the authors need to binarize $\alpha_{i,j}$ to $I_{i,j}$. I think it is okay to make $O_{i,j}'= \alpha_{i,j}*O_{i,j}$, since they do not need to care about the efficiency of the discriminator, unlike the traditional NAS methods. In this way, they could make the discriminator more flexible and do not need to employ STE for the gradient estimation. Also, they do not need to set the threshold $\tau$.

  * I think the authors need to provide some ablation study on their global coordination constraint $L_{global}$. It is hard to tell whether only their selective activation discriminator works, only their $L_{global}$ works, or both work together. It is also possible that $L_{global}$ works alone with the original discriminator without the selective activation.

* Clarity:

  * The paper is overall well-written, but I am confusing about the following two points:

    * The author may need to state how to select a good threshold $\tau$.

    * I think the statement of $L_{local}$ and $L_{global}$ is unclear. The authors may need to explain what is "local" and what is "global". In my understanding, they are just trying to match the loss difference of the teacher and student pair. In this way, there should not be absolute value signs in Equation (6) and (7), which means $L_{global}$ should be
      $$
      L_{global} = |(L_G^S-L_{D_{fake}}^S) - (L_G^T-L_{D_{fake}}^T)|.
      $$
      Moreover, the authors also need to verify that when the student pair have the same loss difference as the teacher pair, the student generator and discriminator maintain the optimal equilibrium.

* Significance: The paper results are good, and I think it could indeed speed up the generator significantly in real-world applications.

**Time Spent Reviewing:**

4

---

> ### Author Response · Authors · 2021-08-10
> **To Reviewer aeqa**
>
> **Q1**: The novelty is somewhat limited. As the authors mention in the related work, Slimmable GAN also tries to maintain the Nash equilibrium by making the discriminator width proportional to the generator width. I think the authors are just applying some NAS methods to automatically prune the discriminator, which is quite similar to Anycost GAN. Although the authors do a good job analyzing the correspondence of different generator and discriminator sizes, the basic idea behind it is not interesting.
>
> **A1**: We analyze the shortcomings of Slimmable GAN and Anycost GAN in lines 99-105 of the original paper. These works merely consider removing the discriminator convolutional channel proportionally or generate the discriminator architecture by GAN loss. However,  our method performs a more fine-grained adaptive task-specific activation strategy for the discriminator.  Rather than simply employ the NAS method to activate the convolutional channels, we analyze the relationship between the loss gap and Nash equilibrium in the introduction section.  And we craftily design two contrast learning constraints into the loss function according to the observed phenomena. Specidically, the local capacity constraint is designed to pursuit capacity balance to maintain the Nash equilibrium and the global coordination constraint helps to pull the loss gap between teacher and student models into close proximity.
>
> **Q2**: I am curious about why the authors need to binarize $\alpha_{i,j}$ to $I_{i,j}$. I think it is okay to make $O_{i,j}'= \alpha_{i,j}*O_{i,j}$, since they do not need to care about the efficiency of the discriminator, unlike the traditional NAS methods. In this way, they could make the discriminator more flexible and do not need to employ STE for the gradient estimation. Also, they do not need to set the threshold t.
>
> **A2**: Because the discriminator contains the Batch Normalization layer.  We binarize$\alpha$ to avoid the situation that the function of $\alpha$ is weakened by Batch Normalization.
>
> **Q3**: I think the authors need to provide some ablation study on their global coordination constraint $L_{global}$. It is hard to tell whether only their selective activation discriminator works, only their  $L_{global}$ works, or both work together. It is also possible that  $L_{global}$ works alone with the original discriminator without selective activation.
>
> **A3**: Thanks a lot for your constructive suggestions and we have conducted additional experiments to address your concerns. Firstly, we directly add $L_{global}$ to the original discriminator without the selective activation discriminator, but the variant only obtains 37.24 mIOU (**decreased by 13.2%**). Secondly, we design a variant that does not employ $L_{global}$ to optimize the selective activation discriminator. It gets the result of 40.21 (**decreased by 6.2%**). The experimental results show that the selective activation discriminator and $L_{global}$ work together to achieve impressive results. We will append these ablation studies to the final version.
>
> **Q4**: The author may need to state how to select a good threshold t.
>
> **A4**: We denote a retention factor $\alpha$ as the retention probability of the corresponding convolution kernel. In this paper, we firstly initialize all retention factors to the constant 1 and limit them to [0, 1]. In the experiment, we choose threshold t = 0.5 based on the probability experience. We also tried other values around 0.5 (such as 0.4, 0.6), but it has little effect on the experimental results.
>
> **Q5**: I think the statement of $L_{local}$ and $L_{global}$ is unclear. The authors may need to explain what is "local" and what is "global". In my understanding, they are just trying to match the loss difference of the teacher and student pair. In this way, there should not be absolute value signs in Equation (6) and (7), which means $L_{global}$ should be
>
> $L_{global} = |(L_G^S-L_{D_{fake}}^S) - (L_G^T-L_{D_{fake}}^T)|$.
>
> Moreover, the authors also need to verify that when the student pair have the same loss difference as the teacher pair, the student generator and discriminator maintain the optimal equilibrium.
>
> **A5**: We constrain selective activation discriminator from two aspects:
> (1) from the local capacity constriant perspective，$L_{local}$ is designed to reduce the loss gap within the student models. (2) from the global coordination perspective, $L_{global}$ adjusts the relative ability of the student pair to be consistent with the teacher pair.
>
> In the experiment (see Figure 1 and 5), we observed $L_G$ is always greater than $L_ D$ is large, so the processing of absolute value is also equivalent to the processing of parentheses.
>
> This global constraint helps to pull the loss gap between teacher and student models into close proximity.  Such restriction does not necessarily guarantee that they maintain the optimal equilibrium, but it can effectively alleviate the problem of excessive loss gap and keep the whole system in the scopes of Nash equilibrium (see Figure 1 and Figure 5 (a)).

---

> > ### Comment · Reviewer_aeqa · 2021-08-30
> > **Thanks for the responses**
> >
> > I appreciate the effort to reply to the comments from all reviews. With the additional explanation and ablation, I am convinced that GCC could maintain a good Nash equilibrium and avoid mode collapse. Both the selective activation discriminator and global constraint $L_{global}$ work together to improve the final performance. However, I still think the novelty is not that enough. Therefore, I only change the score to "6: Marginally above the acceptance threshold". Besides, I also recommend changing $L_{global}$ to $|(L_G^S-L_{D_{fake}}^S) - (L_G^T-L_{D_{fake}}^T)|$, even though $L_G$ is usually larger than $L_{D_{fake}}$, since you could not guarantee $L_G \ge L_{D_{fake}}$ holds for all datasets and during the whole training process.

---

> > > ### Author Response · Authors · 2021-09-01
> > > **Thanks for the responses**
> > >
> > > Thank you for your constructive comments and efforts, we will work hard to modify our paper to make it more perfect. We will adopt your suggestion to revise and discuss the absolute value signs in Equations (6) and (7)  in the final version.

---

### Official Review · Reviewer_sJWF · 2021-07-16

**Rating:** 5
**Confidence:** 4

**Summary:**

In this paper, the authors revisit the role of discriminator in GAN compression and propose a selective activation discriminator to automatically activate the convolutional channels. An online collaborative distillation is adopted to guide the optimization process of the lightweight generator. Experiment results demonstrate the effectiveness of the proposed method.

**Limitations And Societal Impact:**

Yes. See Main Review.

**Main Review:**

Strengths
+ The idea of generator-discriminator cooperative compression scheme for GAN compression is straightforward and reasonable since the balance between generator and discriminator is crucial for GAN training.

+ The importance of discriminator’s channel-number is verified via corresponding experiments, which gives some insights for the image generation community.


Weaknesses
-	My main concern is the lack of novelty. In my view, the proposed mechanism of automatically activating convolutional channels and distillation method are incremental. In addition, the claim that “first attempt …” in line 59 is open to discussion. Some NAS-GAN related works [1][2] also consider the mutual balance between generator and discriminator and search the architectures of both networks simultaneously. The compression and NAS task are essentially the same in my opinion.

-	The ablation studies on table 3 show that the effect of each component is not obvious. Compared with other compression methods, the improvement is also not significant.

-	More analysis about the role of discriminator in GAN Compression should be given since the title of this paper is “Revisiting Discriminator in GAN Compression”. Simply discuss the effect of channel-number is not deep enough. Authors are encouraged to provide more insights.



[1] Gong et al. AutoGAN: Neural Architecture Search for Generative Adversarial Networks. In ICCV. 2019.

[2] Gao et al. AdversarialNAS: Adversarial Neural Architecture Search for GANs. In CVPR. 2020.

---- Update ----

The authors address some of my concerns. However, the novelty of the proposed methods (e.g., selective activation discriminator and global constraint) is still limited in my view, and this work only discusses the effect of channel-number. After reading other comments and author's response, I prefer to keep the original rating 5.

**Time Spent Reviewing:**

4

---

> ### Author Response · Authors · 2021-08-10
> **To Reviewer sJWF**
>
> **Q1**: My main concern is the lack of novelty. In my view, the proposed mechanism of automatically activating convolutional channels and distillation method are incremental. In addition, the claim that “first attempt …” in line 59 is open to discussion. Some NAS-GAN related works [1][2] also consider the mutual balance between generator and discriminator and search the architectures of both networks simultaneously. The compression and NAS task are essentially the same in my opinion.
>
> [1] Gong et al. AutoGAN: Neural Architecture Search for Generative Adversarial Networks. In ICCV. 2019.
>
> [2] Gao et al. AdversarialNAS: Adversarial Neural Architecture Search for GANs. In CVPR. 2020.
>
> **A1**: Our work is to revisit discriminator in the field of GAN compression. In essence, NAS is a process, and model compression is an objective, which are two different concepts. AutoGAN and AdversarialNAS pay more attention to the naive NAS method to obtain the network structure and boost the final results, which fails to take model compression into account.
>
> This setting in our paper is quite different from NAS and provides more task-spefical (i.e., GAN compression) insights. In this paper, we found out the essential reason for the performance degradation in the compression process from the experimental phenomenon and customized the contrast learning constraint to learn the partially activated discriminator. We also introduce a distillation technique to improve the compression performance of the whole framework. And a global coordination constraint is employed to adjust the relative ability of the student generator and discriminator to be consistent with that of the teacher combination.
>
> It is worth noting that distillation is not incremental. The online collaborative distillation is novel to take full advantage of the intermediate feature of the teacher generator and discriminator to further boost the performance of the lightweight generator. Ablation study also proves that this distillation scheme is very effective in GAN compression.
>
> In short,  our method is more intuitive and effective in the GAN compression-oriented scene.
>
> **Q2**: The ablation studies in Table 3 show that the effect of each component is not obvious. Compared with other compression methods, the improvement is also not significant.
>
> **A2**: As shown in Table 3,  each distillation component is effective for improving performance compared with the variants without distillation. Although the improvement of some components is not obvious, it is still very helpful to the total performance. Overall, the model with all components can bring an improvement of 3.71 mIoU (**9.47% improvement**).
>
> As shown in Table. 1 and 2,  our proposed GCC can effectively compress the diverse types of GAN (such as SAGAN, CycleGAN, Pix2Pix, and SRGAN ) with a higher compression rate and almost no sacrifice image fidelity.
>
> For example, GCC compressed more than 22.4% FLOPs and still obtain an improvement of 2.35 mIoU (**5.8% improvement**)  in the Pix2Pix model compared with DMAD. Although GCC compresses CycleGAN with a 95.77% compression rate, it also contributes to reducing FID by 3.65  (**5.8% improvement**) compared with the similar compression rate method, i.e., DMAD.
>
> **Q3**: More analysis about the role of discriminator in GAN Compression should be given since the title of this paper is “Revisiting Discriminator in GAN Compression”. Simply discuss the effect of channel-number is not deep enough. Authors are encouraged to provide more insights.
>
> **A3**: In this paper, we examine the role of discriminators in the field of GAN compression. Through the analysis of the introduction, we know that Nash equilibrium has a key impact on performance in the GAN compression stage. Hence we explore the capacity and guiding role of the discriminator, which contributes to bringing benefits in two aspects. On the one hand, we reduce the discriminator capacity via a selective activation strategy to choose the activated channels.  From the perspective of maintaining performance during compression,  we design a teacher-guided constraint strategy and distill knowledge to guide the learning of the lightweight generator.
>
> The final purpose of GAN compression is to obtain the generator.  The discriminator plays an auxiliary role and will be discarded after the compression stage. So we do not employ some time-consuming way (such as NAS) to compress the discriminator, we focus on a simple but effective constrain dimension (i.e., channel number) in the paper. We will also explore more diverse dimensions of the discriminator in the time-efficient setting in future work.

---

### Official Review · Reviewer_fP8T · 2021-07-16

**Rating:** 6
**Confidence:** 3

**Summary:**

This work introduces a GAN compression method that exploits the role of the discriminator. The main contributions are two-fold: (i) maintain the capacity balance between the compressed generator and discriminator by learning to selectively activate the convolutional channels of the discriminator, (ii) promote the performance of the compressed generator by distilling knowledge from the teacher generator and discriminator to the student models in the feature space. The method is validated on synthesis, translation, and super-resolution tasks.

**Limitations And Societal Impact:**

The limitations discussed in the supplementary material are not convincing. They look more like the general limitations of the topic rather than the proposed method.

**Main Review:**

Originality

+ To my knowledge, the idea is novel, simple, and intuitive. It may potentially open up a new perspective for GAN compression.

+ Nevertheless, it's not the first paper exploiting/distilling the knowledge of the discriminator after training. Several recent works have explored this direction [1-4] for improved sampling. It would be good to see a discussion about that in the related work section and place the work in a broader context of GANs.

Quality

* The proposed method looks technically sound. Both ideas of capacity balance and feature distillation are highly intuitive and well supported by the empirical results.

* However, I'm not fully convinced by the concrete solution to capacity balance. It's known that balancing the learning of G & D is not trivial and it's common in practice that $L_G + L_D$ is not close to 0. Yet, in Eq.6 & Eq.7, the authors propose to measure and balance the capacity by using the value of $L_G$ & $L_D$. I wonder how accurate this is as an approximation of the model capacity. Also, how is it applied to other forms of adversarial loss? It would be good to see a discussion about the potential limitations.

Clarity

* The paper is overall well-motivated and easy to follow.

* Yet, I'm sure not if I fully got the details of online collaborative distillation. E.g., in L182, 'The online scheme optimizes the teacher model and the lightweight student model iteratively and progressively from scratch. Therefore, we do not need a pre-trained teacher model, and directly complete the distillation process in one stage.' Does it refer to learn light-weighted GANs from scratch as opposed to compression?



[1] Discriminator Rejection Sampling, ICLR'19

[2] Metropolis-Hastings Generative Adversarial Networks, ICML'19

[3] Collaborative Sampling in Generative Adversarial Networks, AAAI'20

[4] Your GAN is Secretly an Energy-based Model and You Should use Discriminator Driven Latent Sampling, NIPS'20

**Time Spent Reviewing:**

4

---

> ### Author Response · Authors · 2021-08-10
> **To Reviewer fP8T**
>
> **Q1**: Nevertheless, it's not the first paper exploiting/distilling the knowledge of the discriminator after training. Several recent works have explored this direction [1-4] for improved sampling. It would be good to see a discussion about that in the related work section and place the work in a broader context of GANs.
>
> [1] Discriminator Rejection Sampling, ICLR'19
>
> [2] Metropolis-Hastings Generative Adversarial Networks, ICML'19
>
> [3] Collaborative Sampling in Generative Adversarial Networks, AAAI'20
>
> [4] Your GAN is Secretly an Energy-based Model and You Should use Discriminator Driven Latent Sampling, NIPS'20
>
> **A1**: The purpose of knowledge distillation and sampling is different. Compared with sampling, knowledge distillation aims to transfer the informative concepts from a larger model to the lightweight one.
>
> The proposed GCC method indeed applies discriminator based distillation, but we provide two novel insights to explore the discriminator information:
>
> (1) we employ a local capacity constraint and a global coordination constraint to guide the learning of the selective activation discriminator. The dynamic equilibrium relationship between the original models provides an additional supervisory signal to choose the activated convolutional channels in the discriminator.
>
> (2) A novel online collaborative distillation scheme is designed to simultaneously employ intermediate features of teacher generator and discriminator to guide the optimization process of the lightweight generator step by step. Furthermore, we replace the pre-trained image classification network with the teacher discriminator to calculate the perceptual loss, which provides more task-specifical information for the lightweight generator.
>
> We will add more discussions about sampling and discriminator-based distillation. We also will explore this method in more diverse GAN scenes and tasks in the final version.
>
> **Q2**: However, I'm not fully convinced by the concrete solution to capacity balance. It's known that balancing the learning of G & D is not trivial and it's common in practice that $L_G+L_D$ is not close to 0. Yet, in Eq.6 & Eq.7, the authors propose to measure and balance the capacity by using the value of  $L_G$&$L_D$. I wonder how accurate this is as an approximation of the model capacity. Also, how is it applied to other forms of adversarial loss? It would be good to see a discussion about the potential limitations.
>
> **A2**: As you mentioned,  keeping the balance of generator and discriminator is non-trivial. Hence we introduce global and local constraints to maintain the dynamic balance.  The compressed generator can get rid of mode collapse with the guidance of these constraints.
>
> From the perspective of quantitative results, the 1/4 channels width generator trained with the original discriminator only gets 32.02 mIOU on Cityscapes dataset. While our proposed GCC improved the performance of the 1/4 channels width generator to 37.53 mIOU (**17.2% improvement**).
>
> In addition to quantitative results, we prove the effectiveness of our method through the curve of the loss function and the generative visualization in Figure 5. As depicted in Figure 5(a), the loss gap between the 1/4 channels width generator and selective activation discriminator is more evenly matched to that of the original model as shown in Figure 1(b). As shown in Figure 5(b), GCC-based lightweight generators can provide higher quality images and avoid the forgery trace.
>
> Our method does not necessarily guarantee that it achieves the approximation of optimal model capacity, but it can effectively alleviate the problem of excessive loss gap and keep the whole system in the scopes of Nash equilibrium.
>
> Furthermore,  the formula definition ( Eq.6 and Eq.7) also reveals that our method has good generalization, and it can be employed in diverse optimization schemes (such as task-oriented adversarial loss) that conforms to the basis GAN loss definitions of Eq.1 and Eq.2.
>
> **Q3**: Yet, I'm sure not if I fully got the details of online collaborative distillation. E.g., in L182, 'The online scheme optimizes the teacher model and the lightweight student model iteratively and progressively from scratch. Therefore, we do not need a pre-trained teacher model, and directly complete the distillation process in one stage.' Does it refer to learn light-weighted GANs from scratch as opposed to compression?
>
> **A3**: The purpose of our proposed GCC algorithm is for GAN compression, which aims to get a lightweight generator while maintains its original performance. Specifically, we first use the one-shot pruning method to compress the original generator and obtain a lightweight generator architecture. Because the compression of the generator breaks the Nash equilibrium between it and the discriminator, we design a selective activation discriminator to adaptively maintain the capacity balance with the lightweight generator. To ensure the performance of the lightweight generator after compression, we further propose an online collaborative distillation to transfers the knowledge from uncompressed GAN to boost the performance of the lightweight generator. We summarize the proposed GCC framework in the Appendix, please refer to Algorithm 1.

---

> > ### Comment · Reviewer_fP8T · 2021-09-01
> > **Thanks for the response**
> >
> > Thanks for the clarification! I am more convinced than before. Nevertheless, given the known challenge in balancing the generator and discriminator during compression, it would be good to see not only the strengths but also the limits of the proposed method. I, therefore, keep my rating to 6, and suggest authors add a more thorough analysis in their paper.

---

> > > ### Author Response · Authors · 2021-09-02
> > > **Thanks for the response**
> > >
> > > Thank you again for your affirmation of our paper. We have discussed the limitations of the proposed method in Appendix E  Border Impact, and we will add more limitation discussion in the final version.

---

### Official Review · Reviewer_Ajkh · 2021-07-19

**Rating:** 7
**Confidence:** 4

**Summary:**

This work revisits the role of discriminator in GAN compression and finds that the capacity mismatch when compressing the generator with a fixed discriminator will lead to model collapse. To this end, this work proposes a generator-discriminator co-operative compression framework to automatically adapt the size of discriminator to maintain the Nash equilibrium with the lightweight generator. The extensive evaluation results validate the superiority of the proposed framework over SOTA GAN compression methods.

**Limitations And Societal Impact:**

The limitations are discussed in the main review section.

**Main Review:**

## Originality and significance
This work identifies the model collapse problem in GAN compression caused by a fixed discriminator which is commonly adopted in previous GAN compression works. The proposed framework can effectively tackle this challenge and achieve new SOTA performance in GAN compression based on extensive experiments. Such simple but effective insights may help the community for future GAN compression techniques.


## Quality and clarity
This paper is logically clear and easy to follow with solid motivating analysis. The proposed framework is technically sound with impressive results. I still have the following questions for this paper:

(1) More discussions about the choice of L_{local} in Eq. (7) are expected. Is there any other potential forms to better maintain the Nash equilibrium, e.g., a weighted sum?

(2) [55] and [11] directly discard the discriminator during GAN compression and only reply on the distillation loss, which can also avoid the model collapse. A potential ablation study is to adopt the distillation loss in Eq. (10) while the student discriminator is discarded.


## After rebuttal

The author responses address my concerns. I will keep the original score, i.e., a clear accept.

**Time Spent Reviewing:**

1.5

---

> ### Author Response · Authors · 2021-08-10
> **To Reviewer Ajkh**
>
> **Q1**: More discussions about the choice of $L_{local}$ in Eq. (7) are expected. Is there any other potential forms to better maintain the Nash equilibrium, e.g., a weighted sum?
>
> **A1**: Thank you for such constructive comments. The design of $L_{local}$ is motivated by our experimental observation in Figure. 1. Namely, the incremental loss gap between the generator and the discriminator will break the nash equilibrium in the adversarial training scheme, which leads to the discriminator effortlessly distinguish the authenticity of the input images.
>
> Therefore, we design a simple but effective L1 loss-based local capacity constraint $L_{local}$ to optimize the discriminator. We select L1 loss because it is more stable and insensitive to outliers. In the discriminator optimization procedure, the generator $L_G^S$ can be used as an auxiliary supervision signal to make the discriminator flexibly adjust the activated structure to pursuit capacity balance.
>
> Furthermore, we are also interested to integrate a more fine-grained loss (such as weighted sum) to maintain the Nash equilibrium. We will leave it in further work.
>
> **Q2**: [55] and [11] directly discard the discriminator during GAN compression and only reply on the distillation loss, which can also avoid the model collapse. A potential ablation study is to adopt the distillation loss in Eq. (10) while the student discriminator is discarded. A potential ablation study is to adopt the distillation loss in Eq. (10) while the student discriminator is discarded.
>
> **A2**: We have conducted the corresponding experiments on the Cityscapes dataset by discarding the student discriminator like [11] and [55] in the previous experiments, which obtains 35.5 mIOU (**with a decline of 17.2%**).  It may be due to the fact that although it avoids the model collapse issue,  it fails to take good advantage of GAN loss to provide more supervision information for the lightweight generator. Furthermore, our algorithm contributes to achieving better performance than the methods [11, 55] with a higher compression rate as shown in Table. 1.

---

### Decision · Program_Chairs · 2021-09-27

**Decision:**

Accept (Poster)

**Comment:**

This paper proposes a generator-discriminator co-training compression method to balance the model capacity of the lightweight generator and discriminator during GANs compression. The reviewers appreciate the idea of balancing generator and discriminator as well as comprehensive results and clear writing, although there are still concerns regarding the technical novelty and the ablation studies. The rebuttal addressed most of the reviewers' concerns and highlighted the difference from prior works (e.g., knowledge distillation for discriminator, NAS for discriminator architecture search). I agree with most reviewers and believe that the paper's contribution is significant enough, and the results are convincing. Therefore, I recommend accepting the paper.